# Honey Bees Prefer Pollen Substitutes Rich in Protein Content Located at Short Distance from the Apiary

**DOI:** 10.3390/ani13050885

**Published:** 2023-02-28

**Authors:** Hamed A. Ghramh, Khalid Ali Khan

**Affiliations:** 1Research Center for Advanced Materials Science (RCAMS), King Khalid University, P.O. Box 9004, Abha 61413, Saudi Arabia; 2Unit of Bee Research and Honey Production, King Khalid University, P.O. Box 9004, Abha 61413, Saudi Arabia; 3Biology Department Faculty of Science, King Khalid University, P.O. Box 9004, Abha 61413, Saudi Arabia; 4Applied College, King Khalid University, P.O. Box 9004, Abha 61413, Saudi Arabia

**Keywords:** honey bee, supplementary diet, pollens, foraging behavior

## Abstract

**Simple Summary:**

Pollens being the primary source of protein, lipids, vitamins, and minerals are vital for bee development and reproduction. A major issue confronting beekeeping is developing strong and healthy honey bee colonies. The possibility of prosperous honey bee colonies depends on an effective pollen substitute especially when pollen supply is scarce during the dearth period. Many beekeepers feed their bees different pollen substitutes with sufficient nutrition throughout the period of inadequate pollen quantity or quality. We delivered four different pollen substitutes (chickpea, maize, sorghum, and wheat flour) and natural pollen to honey bee colonies for comparison. Each flour was mixed with a small quantity of cinnamon powder, turmeric powder, and both powders. Further, to investigate the preferred pollen foraging distance from the hives, the best performing pollen substitutes were placed at various distances of 10, 25, and 50 m from the apiary. Chickpea flour (comparatively rich in protein content) located very close to the apiary was the best pollen substitute among the tested flours. This study is very helpful for beekeepers in supplementing their bee colonies when there is a shortage or unavailability of pollens, and it is much better to keep the food source near the apiary.

**Abstract:**

The availability of floral resources is crucial for honey bee colonies because it allows them to obtain protein from pollen and carbohydrates from nectar; typically, they consume these nutrients in the form of bee bread, which has undergone fermentation. However, the intensification of agriculture, urbanization, changes to the topography, and harsh environmental conditions are currently impacting foraging sites due to habitat loss and scarcity of food resources. Thus, this study aimed to assess honey bee preference for various pollen substitute diet compositions. Bee colonies perform poorly because of specific environmental problems, which ultimately result in pollen scarcity. Pollen substitutes located at various distance from the bee hive were also investigated in addition to determining the preferences of honey bees for various pollen substitute diets. The local honey bee (*Apis mellifera jemenitica*) colonies and different diets (four main treatments, namely, chickpea flour, maize flour, sorghum flour, wheat flour; each flour was further mixed with cinnamon powder, turmeric powder, flour only, flour mixed with both cinnamon and turmeric powder) were used. Bee pollen was used as a control. The best performing pollen substitutes were further placed at 10, 25, and 50 m distances from the apiary. Maximum bee visits were observed on bee pollen (210 ± 25.96) followed by chickpea flour only (205 ± 19.32). However, there was variability in the bee visits to the different diets (F (16,34) = 17.91; *p* < 0.01). In addition, a significant difference in diet consumption was observed in control (576 ± 58.85 g) followed by chickpea flour only (463.33 ± 42.84 g), compared to rest of the diets (F (16,34) = 29.75; *p* < 0.01). Similarly, foraging efforts differed significantly (*p* < 0.01) at the observed time of 7–8 A.M., 11–12 A.M., and 4–5 P.M. at the distance of 10, 25, and 50 m away from the apiary. Honey bees preferred to visit the food source that was closest to the hive. This study should be very helpful for beekeepers in supplementing their bee colonies when there is a shortage or unavailability of pollens, and it is much better to keep the food source near the apiary. Future research needs to highlight the effect of these diets on bee health and colony development.

## 1. Introduction

Honey bees (*Apis mellifera jemenitica*) have a high commercial value for honey production [1] as well as pollination in a range of agricultural crops [2]. Honey bees, like other invertebrates, are poikilothermic; they cannot regulate their body temperature and must go into hibernation when the ambient temperature is too high [3]. Due to restricted foraging activity, their dietary requirements and metabolic activities are decreased during this period [3].

High summer temperatures and dry weather are the main factors contributing to honey bee mortality in Saudi Arabia. This is due to decreased plant flowering and pollen availability due to heat stress [4]. Native populations of *A. mellifera jemenitica* in Saudi Arabia are much more tolerant of heat than the common races. *A. mellifera jemenitica* exists in central Saudi Arabia, which has the warmest summer temperatures, >45 °C [5,6]. Additionally, *A. mellifera jemenitica* possesses a remarkable ability to hunt for pollen and possesses high fecundity [7]. The pollen quantity was discovered to have a positive correlation with temperature and a negative correlation with rainfall, relative humidity, and wind speed. The months with the highest pollen quantity (95% of all pollen) were May through September [8] in Abha Saudi Arabia, while in September and October, there was a spike in flight activity. August through September had seen the highest concentrations of pollen brought back to the hive, while November through December had seen the lowest pollen concentrations in the Al-Ahsa region of Saudi Arabia [9].

In addition, other seasonal and climatic fluctuations (precipitation, hail, etc.) produce considerable losses in floral resources throughout the year [10]. Meanwhile, flowers are essential for honey bee brood production, immunological function, and overwintering survival [11,12]. While nectar is a source of carbohydrates, pollen supplies proteins, lipids, and micronutrients [13]. When the natural flora is insufficient, the queen bee’s egg-laying level decreases, resulting in a fall in the colony’s population level [14]. Malnutrition reduces individual survival rates, causes larval life to cease, renders the colony prone to disease, and drives individuals to leave the colony [15,16]. Usually, a honey bee colony obtains 10–26 kg of pollen each year from flowers [17] as a rudimentary source of protein content and amino acid composition for the well-being of their colony [18]. Furthermore, appropriate protein and carbohydrate stores in the colony are suggested to aid honey bees in fighting or tolerating different stressors associated with modern apiculture [17]. Although pollen remains the most desirable and appealing protein source for honey bees, pollen replacements have advantages. Pollen introduced to the colonies from the outside is costly to get in large quantities, and it also entails the danger of introducing infections [19,20] or pesticides [21] into the colonies.

Thus, human intervention is needed to overcome these problems, particularly for disease management and additional feeding. To compensate for the lack of nutritive forage in the environment, hives are routinely given artificial “pollen substitute” diets [22]. As a result, better colony health for honey production and pollination can be maintained [23,24]. To compensate for insufficient pollen forage and boost colony vigor prior to pollination services, beekeepers provide different “pollen substitute” diets [25,26]. In order to manage honey bee colonies during a pollen-scarce season, Pande and Karnatak [27] utilized germinated pulses as a substitute for pollen. For the production of four distinct diets—ger horse gram, ger chickpea, ger green gram/mungbean, and ger pea—various germinated pulse flours were used. Similarly, Kumar and Agrawal [28] made six distinct combinations of artificial food using defatted soy flour, brewer’s yeast, parched gram, spirulina, skim milk powder, sugar, glucose, protein hydrolysate powder, and natural pollen. This artificial diet favored the biochemical composition and net consumption, and also had a positive effect on colony parameters such as egg laying and brood production.

In addition, in another study, different supplements were used such as roasted chickpea flour, broadbean flour, maize flour, and soy flour [29], and these supplements enhanced brood production and longevity. A study conducted in India using four flours—soybean, wheat, maize, and gram—as pollen substitutes found that pollen substitute is crucial for the growth and development of bee colonies not only during times of scarcity but especially during foraging and pollination and to overcome pesticide exposures [30]. Meanwhile, in another study, six protein-rich ingredients—defatted soybean flour, chickpea flour, maize flour, wheat germ, pea flour, and dried brewer’s yeast—were combined in various ratios with sugar powder, bee honey, and water to create ten diets [31], and these diets increased biological activities including diet consumption, sealed worker brood area, and pollen and honey store area.

Honey bees can visit multiple food sources at once and travel up to 11 km to obtain primary food resources such as nectar and pollen [32], which are stored in their colonies as honey and beebread [33]. Their foraging is one of the most well-organized behaviors found in social insects [34]. Honey bee foragers use information gathered from their own experience, such as recall of time and place, sugar concentration to determine whether to continue or begin foraging on certain resources [35]. Until now, research has concentrated on commonly foraged feed elements such as pollen and nectar rather than atypically foraged materials that may be ingested under drought. The objectives of the present study were to evaluate the preference of honey bees for different diets. In addition, we assessed honey bee preferences for various diet supplements placed at various distances from the colonies.

## 2. Materials and Methods

The study was conducted at the Unit of Bee Research and Honey Production, King Khalid University Abha Saudi Arabia. The current study used local honey bee (*A. mellifera jemenitica*) colonies housed in the Langstroth hives. The honey bee colonies placed at the apiary did not show any clinical illness signs (see Figure 1). All bee colonies were subjected to regularly suggested colony management procedures [36].

### 2.1. Preparation of Diets

These pollen substitution diets were high in protein, carbohydrates, minerals, and fats. These items were reasonably priced in the local market. The supplemental diets listed below were created. Each diet was tested with three replications for five pollen substitute diets, including naturally collected pollen as a control on diet preference (see Table 1).

The various supplemental diets were prepared separately first, measured known quantity (see Figure 2), and carefully blended in a dough machine (Hobart dough mixer, model A200, Offenburg, Germany). The flour was fed externally to provide bees easy access to it. Honey bees must vibrate their body to collect powdered substances, a simple process requiring little time and effort [37].

### 2.2. Bee Visits and Diet Consumption

Every day in the late afternoon, the consumption rate of each diet replicate was calculated by measuring diet weight before and after feeding in grams. At the end of the trial, the total amount of food consumed in each replicate was also calculated.

### 2.3. Choice Powder Feeding

This modified procedure used three distances from colonies: 10, 25, and 50 m. Experimental colonies received all five feeds (i.e., CPCM, CPTM, CPOY, CPBH, and PNOY) in separate plates positioned at various distances [37] as indicated in Figure 3.

### 2.4. Estimation of Honey Bee Numbers

The numbers of honey bees were counted at different time intervals (7–8 A.M., 11–12 A.M., and 4–5 P.M.). During each time interval, the number of honey bees were counted visually three times as indicated in Figure 3.

### 2.5. Diet Consumption

The diet weight between before and after feeding in grams per colony were calculated to determine the net weight of pollen-supplemented diets ingested within treatments after feeding 10 times to each colony [38].

### 2.6. Statistical Analysis

The total amount of preferred food consumed and distance preference was compared between treatments with an analysis of variance. The data were calculated as mean and standard error using the SPSS (version 20). Graphs were created with the GraphPad Prism software (version 7.03). Furthermore, the Tukey post hoc test was used for multiple group comparisons at the 0.05 level.

## 3. Results

### 3.1. Honey Bee Visitation

Maximum honey bee visitation during the first week was on the PNOY with 143 ± 12.89/week followed by CPOY (133.67 ± 13.75/week) and CPBH (78 ± 38.07/week). The fewest visits were on the MZOY (1.33 ± 0.88/week) followed by SGOY and WTOY with 2 ± 0.57/week and 2 ± 1.52/week mean visits, respectively. In the second week, maximum honey bee visits were observed in the CPOY (71.66 ± 6.06/week) and PNOY (67 ± 13.11/week). The fewest visits were on the SGOY (0.33 ± 0.33/week) followed by WTOY (0.66 ± 0.66/week) and MZOY (1 ± 0.57/week). However, overall time periods, maximum visits were on the PNOY (210 ± 25.96) followed by CPOY (205 ± 19.32), and the fewest visits were on the MZOY and SGOY with mean visits of 2.33 ± 1.45 and 2.33 ± 0.88 followed by WTOY (2.66 ± 2.18) as indicated in Figure 4.

### 3.2. Diet Supplement Consumption

The findings showed that honey bees ingested varied amounts of all supplements throughout the research period. Honey bees consumed the highest amount of PNOY diet (404 ± 28.15 g/week) in week 1, which was followed by CPOY (350 ± 8.21 g/week) and CPBH (235.67 ± 7.85/week). The least consumption was for the WTOY diet (5 ± 4.04 g/week) followed by MZOY (7.66 ± 4.97 g/week) and SGOY (11 ± 2 g/week) as indicated in Figure 2. During week 2, the highest diet consumption amount was observed for the CPTM diet (203.33 ± 16.41 g/week) followed by the CPBH diet (192 ± 6.42 g/week), and then PNOY (172 ± 38.15 g/week). However, the overall highest consumption was for the PNOY diet (576 ± 58.85 g) followed by the CPOY diet (463.33 ± 42.84 g), the CPBH diet (427.67 ± 6.35 g), and the CPTM diet (384.67 ± 14.72 g). The least consumption was observed for the WTOY diet (7 ± 6.02 g) followed by the MZCM (16 ± 8.32) and MTTM diets (63 ± 21.37). The remaining diet treatment consumption is indicated in Figure 5.

ANOVA was performed on the honey bee visits and diet consumption, and there were significant differences between the visits to the different treatments during the first week (*p* < 0.01). Similarly, during the second week, there were significant visits to different diets (*p* < 0.01). Overall, there were significant variability in some of the treatment visits (*p* < 0.01) as indicated in Table 2.

According to Table 2, the mean diet consumption throughout the entire sample of honey bee colonies showed notable variability during the first week (*p* < 0.01). The same pattern was observed in consumption for the second week (*p* < 0.01). Similarly, overall, there were significant differences in diet consumption due to treatment (*p* < 0.01).

There was a strong positive correlation between the two variables: honey bee visits and diet consumption. During the first week, the Pearson’s correlation was 0.957. Similarly, during the second week and in the total observation period, a strong positive correlation was also found between bee visits and consumption: 0.89 and 0.94, respectively.

### 3.3. Foraging Efforts

The foraging effort was measured by counting the number of bees visiting the best performing diet (chickpea flour mixed with different spices) placed at different distances such as 10 m, 25 m, and 50 m and observed at different time intervals. The maximum mean number of honey bee visits during 7–8 A.M. at a 10 m distance was observed in the PNOY diet (282.5 ± 2.5), which was followed by CPOY (277.5 ± 7.5). The visitation rate was observed in CPCM, which was 75 ± 5, followed by CPBH (79.5 ± 0.5). The maximum mean number of honey bees visiting the PNOY diet (145.5 ± 3) at 25 m was less than the maximum visitation rate to CPOY (252.5 ± 2.5). The least number of visits were observed in the CPTM diet with 50 ± 5 followed by the CPBH diet (51 ± 1). Similarly, at the distance of 50 m, maximum number of honey bees were observed in the PNOY diet (237 ± 7) followed by the CPOY diet (215.5 ± 14.5). The least number of visits (34.5 ± 4.5) were recorded at the CPCM diet followed by the CPBH diet (37.5 ± 2.5) as shown in Figure 6a.

As indicated in Figure 6b, when diets were placed at the distance of 10 m, the maximum number of honey bees during the time of 11–12 A.M. were observed on the PNOY (226 ± 1) followed by CPOY (220.5 ± 0.5) while the fewest number of honey bees were recorded on the CPBH diet, which was 58.5 ± 0.5 followed by CPCM (65 ± 5). Similarly, when diets were placed at a distance of 25 m, the maximum number of honey bees visited the COPY diet (202.5 ± 6.5) followed by the PNOY diet (202 ± 3), while the fewest number of visits were observed on the CPCM diet with 35 ± 5 visits followed by the CPBH diet (46.5 ± 5.5). In the case of 50 m, maximum visits were observed on the copy diet (180 ± 10) followed by the PNOY diet (176 ± 1). The least number of visits were recorded on the CPCM and CPBH diets (27.5 ± 2.5 and 40.5 ± 5.5, respectively).

When the foraging activity of honey bees at the time of 4–5 P.M. was observed at a distance of 10 m, the maximum number of honey bees visiting the diets were found on the PNOY treatment (205 ± 10) followed by the copy treatment (178 ± 7). The fewest bees were observed on the CPCM (50 ± 5) and the CPBH diets (51 ± 1). At the distance of 25 m, maximum number of bee visits were observed on the PNOY diet (185 ± 10) followed by the copy diet (164 ± 6). The least number of honey bee visits were recorded on the CPCM (32.5 ± 1.5) and the CPBH diets (37 ± 8). When diets were placed at a distance of 50 m, the maximum number of honey bees visiting the treatments were found on the PNOY diet (145 ± 9), followed by the copy diet (140 ± 10). The least number of honey bees visiting a diet were found on the CPCM (22.5 ± 2.5) and the CPBH diets (26 ± 3) as indicated in Figure 6c.

There was a non-statistically significant interaction between time of day, distance from the colonies, and diet types (F (16,45) = 1.017 (*p* = 0.458). However, a significant interaction was found between time of day and distance (F (16,45) = 3.063 (*p* = 0.026). Similarly, a significant interaction was found between time of day and diet types (F (16,45) = 33.349 (*p* = 0.001).

## 4. Discussion

Like most other invertebrates, honey bees are poikilothermic; they are unable to control their body temperatures and become inactive when the outside temperature becomes intolerable. Due to severely constrained foraging activities during hot weather, their nutritional needs and metabolic activity are reduced [3]. Thus, the present study was designed to offer a substitute for honey bees in the harsh conditions of KSA and to investigate the effects of distance from the bee hive on visitation rate to the diets.

Overall, in our study, honey bees exhibited a significant difference in visitation rates to the different diets. However, maximum visits were observed in PNOY (pollen as a control) (210 ± 25.96) followed by CPOY (chickpea flour only) (205 ± 19.32). In terms of diet consumption, PNOY (576 ± 58.85 g) followed by CPOY (463.33 ± 42.84 g) were maximum diets consumed which are in agreement with findings of Khan and Ghramh [39]. They concluded that honey bees ingested much more pollen (11.51 ± 2.22 mg/bee) and ajeena diet, i.e., commercially available pollen substitute (10.68 ± 1.29 mg/bee) than any other diet. This attraction towards pollen may be due to the quality of the diet; pollen attracts foragers [40] and is considered a major source of vitamins, minerals, lipids, carbohydrates, sterols, proteins, and amino acids [41]. The nutritional value of protein would be the primary factor in honey bees’ selection of pollen for food [42]. In a study conducted in India, four distinct germinated pulse flour diets—germinated horse gram, germinated chickpea, germinated green gram/mungbean, and germinated pea—were used and following feeding, foraging behavior was seen in all diet combinations, including germinated chickpea, germinated green gram, and germinated horse gram [27]. While our findings were in contrast with the study conducted in India, three diet formulas using the four germinated pulses soybean, mungbean, pigeon pea, and chickpea were used, and soybean was the most preferred of the four pulses and three formulations [43]. However, the other substitutes can also be very helpful in drought or harsh conditions when there are few flowers. The fluctuation in floral sources and bee colony population density affect the annual pollen supply for bee colonies in many regions of the world. Since the flora that honey bees need is not consistently present, artificial pollen substitutes and supplements have been utilized to sustain the strength of bee colonies by lengthening the adult lifespan and maintaining brood area [44]. While these supplemented meals and pollen replacements may offer a temporary solution to avoid bee losses in poor foraging settings, it cannot be sustained as a long-term solution in a pollen scarce locale.

In our study, there were reasonable numbers of honey bees that visited and consumed chickpea flour only (CPOY). This can be the best option when there is a pollen shortage. This is because chickpea flour has a good amount of protein (21.70–23.70%), carbohydrates (59.66–66.42%), fats (4.80–6.36%), ash (2.2–3.46%), total fiber (14.80), and moisture contents (9.35%) [45,46,47]. There are also other flours that have high amounts of proteins and other important elements (Table 3).

In the present study, when honey bee visits were compared at different timings (7–8 A.M., 11–12 A.M., and 4–5 P.M.), and also diets were placed at different distances (10, 25, and 50 m) from the bee hive, maximum activity was recorded in the morning (7–8 A.M.) and at the closest distance of 10 m from the hive. In another study, contrasting results were reported. Honey bees spent the most time foraging during the day, particularly at 12:00 P.M., followed by 14:00 P.M., and finally at 10:00 A.M. every week [57]. Similar to these findings, Pernal and Currie [58] found that honey bees foraged more frequently in the afternoon than in the morning. Forager bees increased their activity and pollen collecting in an onion crop between the hours of 11:00 A.M. and 12:00 P.M. over several days [59]. In all of these studies, maximum bee foraging activity was in the afternoon. This may be due to the effect of rising temperatures and falling relative humidity on anther dehiscence, reaching their highest from 11:00 A.M. to 14:00 P.M. This time period corresponded to the peak pollen-gathering activity of the bees [60]. However, in our case, pollens were used as a control diet detached from flowers and that may be the reason why honey bees showed maximum activity in the morning (7–8 A.M.) instead of the afternoon due to the free access to pollen early in the morning. In our study, there was a constant increase in the percentage of bees foraging for pollen in the morning, but in the afternoon, that percentage declined. It was discovered that many honey bees flew orientation flights between 12:30 and 14:00 h, especially in sunny conditions, which led to a decreased percentage of pollen foragers in colonies during the middle of the day. This was also documented by a study in China [60]. Additionally, our findings revealed a substantial decrease in foraging activity from 12:00 h to 4–5 P.M., which was likely caused by the high air temperature (exceeding 40 °C). High ambient temperatures make foraging more energy-intensive and can lead to dehydration of foragers. Importantly, foraging distance varies with month differently for the two kinds of forage. In some months, we observe greater distance for one forage type and other months we see the opposite. Overall, this implies that the distance that foraging honey bees must travel is not greatly influenced by one type of forage over another, with summer generally being the season where bees must go farther to gather forage than spring or fall. In the present study, there was a strong positive correlation between the visits and diet consumption (*p* < 0.01); as the visits increased, the amount of diet consumed also increased. Hence, our results suggest that when food sources are abundant near the bee hive, consumption and foraging activity may increase.

Additionally, more field research is required to ascertain how these supplemental diets affect the health and productivity of honey bee colonies. This research could assist beekeepers in creating more suitable food products that reduce waste and improve the nutritional intake of their bee colonies, especially in harsh conditions when there is a shortage of flora, particularly in the case of the KSA region.

## 5. Conclusions

In the present study, we examined the preference of honey bees towards different diets and observed the effect of distance on foraging behavior. Overall, honey bees were more attracted to the natural pollens than the other diets. However, this does not mean that honey bees were not attracted to other diets; there were a reasonable number of honey bees that visited the alternative diets such as the chickpea flour only diet. These supplemental diets can be very helpful when there is a scarcity of pollens, and they can play a very important role in the production of honey. In terms of distance, honey bees preferred to visit the food source nearest to the bee hive (10 m), and the preferred time was in the morning (7–8 A.M.). More research is required to learn how these supplements affect different physiological parameters of honey bee races under diverse climatic situations.

## Figures and Tables

**Figure 1 animals-13-00885-f001:**
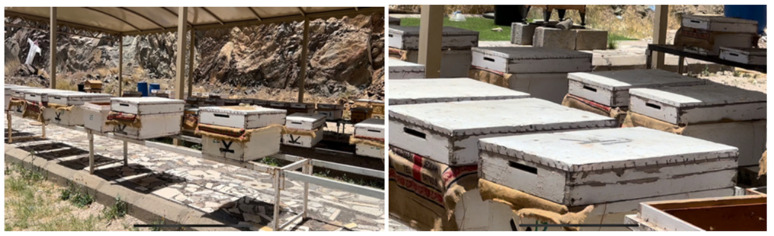
*A. mellifera jemenitica* apiary set-up.

**Figure 2 animals-13-00885-f002:**
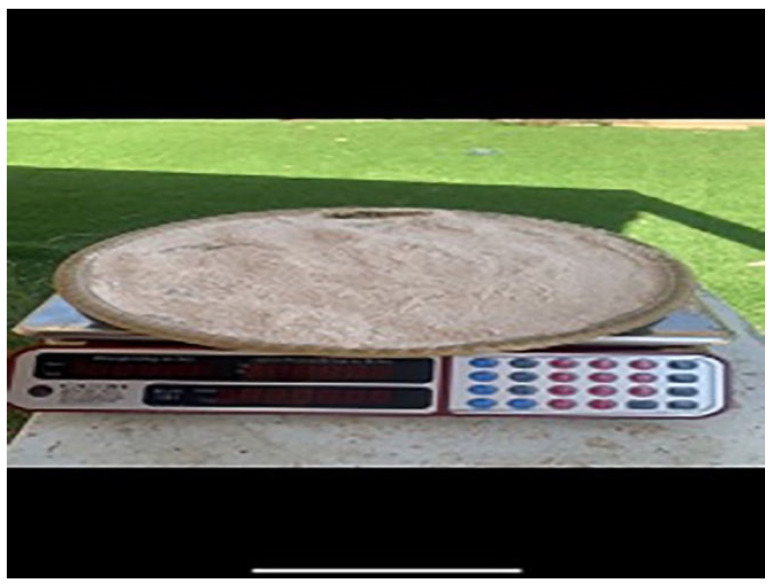
Measuring known quantity of diets before mixing.

**Figure 3 animals-13-00885-f003:**
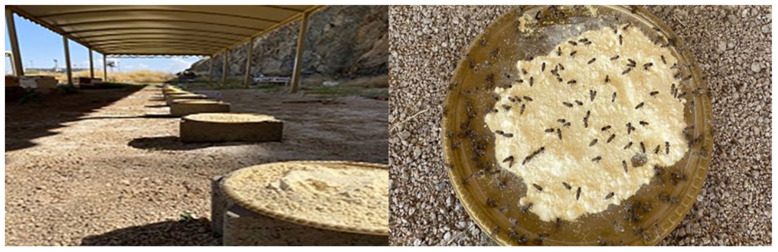
Different diet supplements at different distances (10, 25, and 50 m) away from the apiary and measuring the number of honey bees.

**Figure 4 animals-13-00885-f004:**
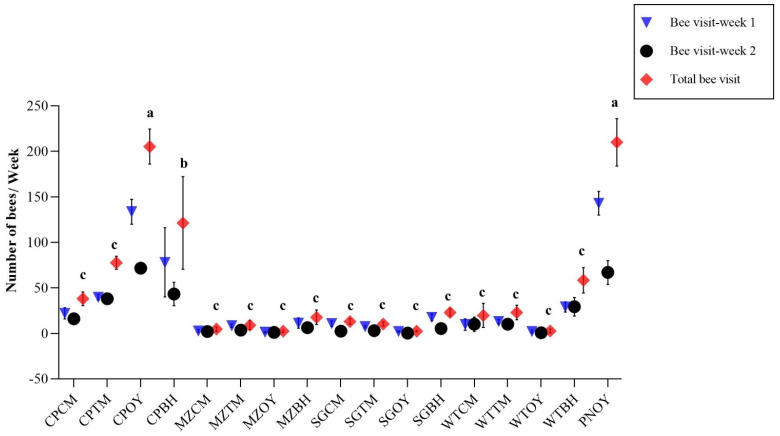
Honey bee visits towards different diet compositions. Whereas, CPCM = Chickpea flour + Cinnamon powder, CPTM = Chickpea flour + Turmeric powder, COPY = Chickpea flour only, CPBH = Chickpea flour + Both powders, MZCM = Maize flour + Cinnamon powder, MZTM = Maize flour + Turmeric powder, MZOY = Maize flour only, MZBH = Maize flour + Both powders, SGCM = Sorghum flour + Cinnamon powder, SGTM = Sorghum flour + Turmeric powder, SGOY = Sorghum flour only, SGBH = Sorghum flour + Both powders, WTCM = Wheat flour + Cinnamon powder, WTTM = Wheat flour + Turmeric powder, WTOY = Wheat flour only, WTBH = Wheat flour + Both powders, PNOY = Pollen only as a control. Different superscript letters indicate significant differences between total diet consumption.

**Figure 5 animals-13-00885-f005:**
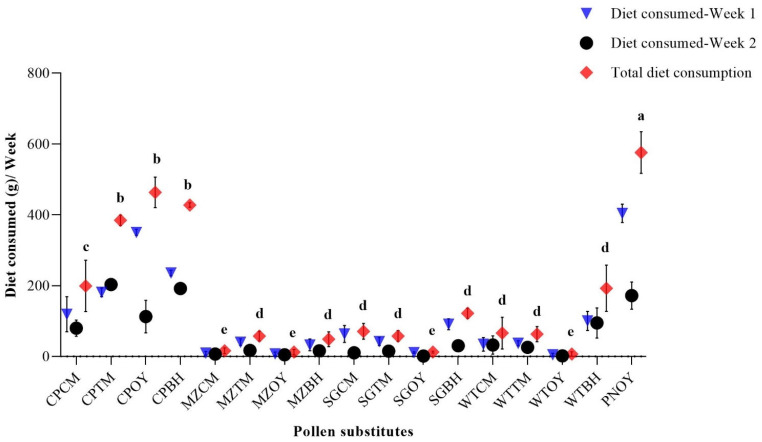
Honey bee diet consumption from various pollen substitutes. Whereas, CPCM = Chickpea flour + Cinnamon powder, CPTM = Chickpea flour + Turmeric powder, CPOY = Chickpea flour only, CPBH = Chickpea flour + Both powders, MZCM = Maize flour + Cinnamon powder, MZTM = Maize flour + Turmeric powder, MZOY = Maize flour only, MZBH = Maize flour + Both powders, SGCM = Sorghum flour + Cinnamon powder, SGTM = Sorghum flour + Turmeric powder, SGOY = Sorghum flour only, SGBH = Sorghum flour + Both powders, WTCM = Wheat flour + Cinnamon powder, WTTM = Wheat flour + Turmeric powder, WTOY = Wheat flour only, WTBH = Wheat flour + Both powders, PNOY = Pollen only as a control. Different superscript letters indicate significant differences between total diet consumption.

**Figure 6 animals-13-00885-f006:**
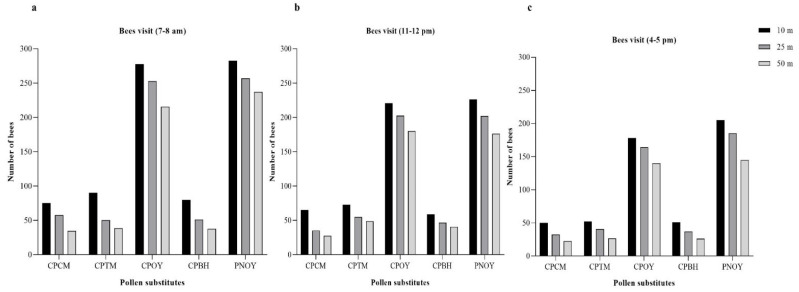
Honey bee visits at different times and pollen substitutes present at different distances: (**a**) 7–8 A.M. time (**b**) 11–12 A.M. time (**c**) 4–5 P.M. time. Whereas, CPCM = Chickpea flour + Cinnamon powder, CPTM = Chickpea flour + Turmeric powder, CPOY = Chickpea flour only, CPBH = Chickpea flour + Both powders, PNOY = Pollen only as a control.

**Table 1 animals-13-00885-t001:** Different supplement formulations.

Sr.#	Pollen Supplementary Diet	Ratio	Abbreviations
1	Chickpea flour + Cinnamon powder	50:1	CPCM
2	Chickpea flour + Turmeric powder	50:1	CPTM
3	Chickpea flour only	-	CPOY
4	Chickpea flour + Both powders	50:1	CPBH
5	Maize flour + Cinnamon powder	50:1	MZCM
6	Maize flour + Turmeric powder	50:1	MZTM
7	Maize flour only	-	MZOY
8	Maize flour + Both powders	50:1	MZBH
9	Sorghum flour + Cinnamon powder	50:1	SGCM
10	Sorghum flour + Turmeric powder	50:1	SGTM
11	Sorghum flour only	-	SGOY
12	Sorghum flour + Both powders	50:1	SGBH
13	Wheat flour + Cinnamon powder	50:1	WTCM
14	Wheat flour + Turmeric powder	50:1	WTTM
15	Wheat flour only	-	WTOY
16	Wheat flour + Both powders	50:1	WTBH
17	Pollen only as a control	-	PNOY

**Table 2 animals-13-00885-t002:** Honey bee visit and diet consumption (ANOVA).

Honey Bee Visits	Sum of Squares	Degrees of Freedom	Mean Square	F	*p*-Value
Week 1	94,945.020	16,34	5934.064	16.883	>0.001
Week 2	26,137.17	16,34	1633.574	14.922	>0.001
Total	218,023.64	16,34	13,626.478	17.913	>0.001
Diet consumption					
Week 1	705,552.03	16,34	44,097.002	42.080	>0.001
Week 2	238,175.92	16,34	14,885.995	12	>0.001
Total	1,615,024.64	16,34	100,939.020	29.758	>0.001

**Table 3 animals-13-00885-t003:** Nutritional content of pollen substitutes reported in different studies.

Pollen Substitutes	Protein (%)	Carbohydrates (%)	Fat (%)	Ash (%)	Total Dietary Fiber	Moisture (%)	References
Wheat flour	10.55	74.88	0.94	0.94	0.36	12.67	[48]
	11.85	86.04	1.06	0.52	0.54	11.97	[49]
	11.60	78.70	1.70	0.97	-	14.20	[50]
Maize flour	6.00	-	2.18	0.61	-	10.63	[51]
	8.90	-	5.30	1.30	15.60	-	[52]
	8.55	78.77	2.61	0.52	3.68	9.55	[53]
Chickpea flour	21.85	66.42	6.36	2.92	-	-	[47]
	21.70	59.66	5.81	3.46	-	9.35	[45]
	23.70	61.10	4.80	2.2	14.80	-	[46]
Sorghum flour	12.30	73.80	3.60	2.92	-	-	[54]
	12.21	83.45	3.76	0.68	-	-	[55]
	11.50	72.00	2.70	-	-	-	[56]

## Data Availability

Data are contained within the article.

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
