# Peer review of "Honey Bees Prefer Pollen Substitutes Rich in Protein Content Located at Short Distance from the Apiary"

_animals, 2023, doi:10.3390/ani13050885_

Round 1

Reviewer 1 Report

The authors investigated the attractiveness of various bee diets in this paper. They conducted several experiments and collected data, but the MS has significant issues.   1. The experimental design is too basic and insufficient for scientific research. This subject may be more appropriate for a scientific-popular magazine or specialized bee literature. More data would be required for a more in-depth investigation. 2. It is impossible to understand how the experiments were carried out and evaluated because the Material and Methods section is written too shoddily and haphazardly. This makes it impossible to comprehend the results. 3. The tested diets are labeled in an extremely complicated manner (abbreviations in Table 1). This is impossible to remember, which makes it very difficult to understand the results. 4. Results section: Figures 1–3 do not say anything and should not be included; Figure 3 appears twice in the text (on pages 4 and 9); and Figure 6 does not contain parts a, b, and c as stated on page 8 (l. 217, 218 and 235); all Figures have insufficient legends

Author Response

Response to reviewer 1

 The authors investigated the attractiveness of various bee diets in this paper. They conducted several experiments and collected data, but the MS has significant issues.   

Comment 1:

The experimental design is too basic and insufficient for scientific research. This subject may be more appropriate for a scientific-popular magazine or specialized bee literature. More data would be required for a more in-depth investigation. 

Response: Thank you so much for your friendly suggestion. No doubt the experiment design is basic but the results are useful for practical beekeepers. The results of this study will definitely help the beekeepers to opt best performing pollen substitute especially in dearth period. This study will also help to place the best performing pollen substitute placed at certain distance for maximum benefits.

Comment 2:

It is impossible to understand how the experiments were carried out and evaluated because the Material and Methods section is written too shoddily and haphazardly. This makes it impossible to comprehend the results.

Response: We have tried to improve the methodology section in the improved version of the manuscript.

 Comment 3: The tested diets are labeled in an extremely complicated manner (abbreviations in Table 1). This is impossible to remember, which makes it very difficult to understand the results. 

Response: Thank you so much for your valuable comments. All Abbreviations are extracted from the name of used pollen substitute.

Comment 4:

Results section: Figures 1–3 do not say anything and should not be included; Figure 3 appears twice in the text (on pages 4 and 9); and Figure 6 does not contain parts a, b, and c as stated on page 8 (l. 217, 218 and 235); all Figures have insufficient legends

Response: Thank you again for your comment. Figures 1-3 help to describe the methodology. That’s why we added these figures. We have corrected the Figure 3 appearance and changed the figure label on page 9.  In addition, we have updated the figures legends in the whole manuscript.

Reviewer 2 Report

Comments and Suggestions:

Your paper is interesting and an important contribution to what is known about supplemental feeding. Your literature review in the Introduction and Discussion is thorough. However, it needs a bit of revising before I think it is acceptable for publication.

1. Please see my edit suggestions for improving the English grammar included in the manuscript.

2. When you describe the ranking of bee visitation to diets and consumption of diets you need to fully state the units of your dependent variables. So, is it visits / week? Visits / hour? And same with consumption, g/week?, etc. Please state the units in the text and in the Figures.

3. In the Figures please include the letters from the post-hoc Tukey comparisons for all 17 diets. In order to avoid clutter just do this for the mean total bee visitation rates and mean total consumption. 

4. When discussing the ranking of BOTH visitation rate and Consumption rate in the text you state which treatments had the highest visitation or consumption and the next highest and then you state the lowest. Please be consistent with your post-hoc tests. Mention if the two or three highest are significantly different from one another. The same for the lowest, mention if they are significantly lower than the highest diets and if there are differences among the lowest diets.

5. Whenever you state F-test results to back up your findings, you need to include BOTH the numerator (treatment) and denominator (error) degrees of freedom for the calculated F-values and P values to have validity. See manuscript text for an example of how to report it.

6. You should delete Table 2 because all of the pertinent information regarding your ANOVAs are in the text. Either delete Table 2, or take out the F-test results from the text and include both degrees of freedom in the table. However, either way, you do not want to repeat the information and have it BOTH in the text and table.

7. Delete Table 3 and report the three correlation coefficients in the text as is suggested in the manuscript.  

8. All figures, make sure that you have the proper units for the dependent variables (y-axes). Also, the Figure numbers are incorrect, see manuscript and correct.

9. For the experiment where you looked at three independent variables for the 5 diets: 1) the time of day, 2) distance from colonies, and 3) diet type. The multiple One-way ANOVAs that you conducted were not a correct way to analyze your data. You NEED to conduct a three-way ANOVA. Once you run the three-way ANOVA first look at the three-way interaction: time of day X Distance X diet type. If it is significant (P<0.05), then there is no need to go any further. Interpret the mean rankings and explain why there is a three-way interaction. If the three-way interaction is not significant, then go to the two-way interactions: 1) Time of day X Distance, 2) Time of Day X Diet type. If any are significant then interpret the pattern and explain why they are significant.  For example, looking at Figure 6 there might be a Time of day X Diet type interaction. There appears to be much less difference between the “preferred” diets COPY and PNOY and the other diets at Time of day 4-5PM compared to times 7-8 AM and 11-12PM. 

BUT if NONE of the interactions are significant then you can look at the significance of the three main effects independently. Do not bother interpreting the main effect IF you have significant interaction effects. Now once you find out which interactions or main effects are significant you can decide how to make Figure 6 so that it best illustrates the significant effects that you plan on discussing. Also, include the standard deviations in the Figure.

10. Again, if you include the F-tests in the text when discussing the Time of day X Distance X Diet type experiment, then you will not need to put the results of the Three-way ANOVA in Table 4.     

My comments are not meant to be harsh. I am trying to help you improve your nice study. Good Luck!

Author Response

Response to reviewer 2

Comment 1:

Please see my edit suggestions for improving the English grammar included in the manuscript.

Response: Thank you so much for a comprehensive review of our manuscript. We have properly followed all your suggestions for improving English grammar.

Comment 2:

When you describe the ranking of bee visitation to diets and consumption of diets, you need to fully state the units of your dependent variables. So, is it visits / week? Visits / hour? And same with consumption, g/week? etc. Please state the units in the text and in the Figures.

Response: We have improved the manuscript as per your suggestion.

Comment 3:

In the Figures, please include the letters from the post-hoc Tukey comparisons for all 17 diets. In order to avoid clutter just do this for the mean total bee visitation rates and mean total consumption. 

Response: We have modified the figures as suggested and included the letters from the post-hoc Tukey comparisons for all 17 diets. Now the figures in revised version look like as given below;

Figure 4. Honey bee visits towards different diet compositions.

Figure 5. Honey bee diet consumption from various pollen substitutes.

Comment 4:

When discussing the ranking of BOTH visitation rate and Consumption rate in the text you state which treatments had the highest visitation or consumption and the next highest and then you state the lowest. Please be consistent with your post-hoc tests. Mention if the two or three highest are significantly different from one another. The same for the lowest, mention if they are significantly lower than the highest diets and if there are differences among the lowest diets.

Response: Done.

Comment 5:

Whenever you state F-test results to back up your findings, you need to include BOTH the numerator (treatment) and denominator (error) degrees of freedom for the calculated F-values and P values to have validity. See manuscript text for an example of how to report it.

Response: We have improved this point in the manuscript as per suggested.

 Comment 6:

You should delete Table 2 because all of the pertinent information regarding your ANOVAs are in the text. Either delete Table 2, or take out the F-test results from the text and include both degrees of freedom in the table. However, either way, you do not want to repeat the information and have it BOTH in the text and table.

Response: Thanks for your excellent comment. We have modified this point as per suggestions.

Comment 7:

Delete Table 3 and report the three correlation coefficients in the text as is suggested in the manuscript.  

Response: Done.

Comment 8:

All figures, make sure that you have the proper units for the dependent variables (y-axes). Also, the Figure numbers are incorrect, see manuscript and correct.

Response: We have corrected the figure's axes and numbered them in the manuscript.

Comment 9:

For the experiment where you looked at three independent variables for the 5 diets: 1) the time of day, 2) distance from colonies, and 3) diet type. The multiple One-way ANOVAs that you conducted were not a correct way to analyze your data. You NEED to conduct a three-way ANOVA. Once you run the three-way ANOVA first look at the three-way interaction: time of day X Distance X diet type. If it is significant (P<0.05), then there is no need to go any further. Interpret the mean rankings and explain why there is a three-way interaction. If the three-way interaction is not significant, then go to the two-way interactions: 1) Time of day X Distance, 2) Time of Day X Diet type. If any are significant then interpret the pattern and explain why they are significant.  For example, looking at Figure 6 there might be a Time-of-day X Diet type interaction. There appears to be much less difference between the “preferred” diets COPY and PNOY and the other diets at Time of day 4-5PM compared to times 7-8 AM and 11-12PM. 

BUT if NONE of the interactions are significant then you can look at the significance of the three main effects independently. Do not bother interpreting the main effect IF you have significant interaction effects. Now once you find out which interactions or main effects are significant you can decide how to make Figure 6 so that it best illustrates the significant effects that you plan on discussing. Also, include the standard deviations in the Figure.

Response: We have performed the three-way ANOVA as suggested. In addition, we have modified the results in the manuscript.

Comment 10:

Again, if you include the F-tests in the text when discussing the Time-of-day X Distance X Diet type experiment, then you will not need to put the results of the Three-way ANOVA in Table 4.     

Response: We have deleted table 4 in the manuscript.

Reviewer 3 Report

The reviewed work undertook a study of honey bee preferences for different diets and observations of the effect of distance on foraging behaviour, in which a reasonable number of honey bees were observed visiting alternative diets with a clear preference for foraging close to the hive. The subject matter of the paper undertaken appears interesting and the conclusions of the study useful for practical application. The manuscript meets the requirements of the journal and the documentation collected seems sufficient. The only question that arises is whether the authors of the paper should not indicate permission to conduct this type of research (issue number and year of the document)?  There is also no statement from the authors regarding potential conflicts of interest in relation to this work (whether or not there is one). Minor shortcomings do not diminish the value of the paper. 

Author Response

Response to reviewer 3

General comments:

The reviewed work undertook a study of honey bee preferences for different diets and observations of the effect of distance on foraging behavior, in which a reasonable number of honey bees were observed visiting alternative diets with a clear preference for foraging close to the hive. The subject matter of the paper undertaken appears interesting and the conclusions of the study useful for practical application. The manuscript meets the requirements of the journal and the documentation collected seems sufficient. The only question that arises is whether the authors of the paper should not indicate permission to conduct this type of research (issue number and year of the document)?  There is also no statement from the authors regarding potential conflicts of interest in relation to this work (whether or not there is one). Minor shortcomings do not diminish the value of the paper. 

Response: Thank you so much for your valuable suggestions. The authors conducted the research at the Unit of Bee Research and Honey Production, King Khalid University, Saudi Arabia and they are affiliated with this institute. Furthermore, the statement regarding potential conflicts of interest in relation to this work is clearly mentioned in revised version of this manuscript according to you suggestion.

Round 2

Reviewer 2 Report

The authors have responded appropriately and deligently to my suggestions and comments. The only minor comment that I have is that in the current Table 2. The column for P values should read ">0.001" instead of ".000" because the probability of an F value only approaches 0 and never becomes 0. 

Other than this comment I believe that this paper is ready to publish.